# Spectral Demodulation of Mixed-Linewidth FBG Sensor Networks Using Cloud-Based Deep Learning for Land Monitoring

**DOI:** 10.3390/s25185627

**Published:** 2025-09-09

**Authors:** Michael Augustine Arockiyadoss, Cheng-Kai Yao, Pei-Chung Liu, Pradeep Kumar, Siva Kumar Nagi, Amare Mulatie Dehnaw, Peng-Chun Peng

**Affiliations:** Department of Electro-Optical Engineering, National Taipei University of Technology, Taipei 10608, Taiwan or pushpamichaeldoss@gmail.com (M.A.A.); t111659004@ntut.org.tw (C.-K.Y.); t113658043@ntut.edu.tw (P.-C.L.); t111999408@ntut.org.tw (P.K.); t1129998403@ntut.org.tw (S.K.N.); amare.mulatie@ntut.edu.tw (A.M.D.)

**Keywords:** fiber Bragg grating, deep learning, sensor networks, optical sensors, land monitoring, agriculture, maritime sensing, urban infrastructure, spectral analysis

## Abstract

Fiber Bragg grating (FBG) sensing systems face significant challenges in resolving overlapping spectral signatures when multiple sensors operate within limited wavelength ranges, severely limiting sensor density and network scalability. This study introduces a novel Transformer-based neural network architecture that effectively resolves spectral overlap in both uniform and mixed-linewidth FBG sensor arrays, operating under bidirectional drift. The system uniquely combines dual-linewidth configurations with reflection and transmission mode fusion to enhance demodulation accuracy and sensing capacity. By integrating cloud computing, the model enables scalable deployment and near-real-time inference even in large-scale monitoring environments. The proposed approach supports self-healing functionality through dynamic switching between spectral modes during fiber breaks and enhances resilience against spectral congestion. Comprehensive evaluation across twelve drift scenarios demonstrates exceptional demodulation performance under severe spectral overlap conditions that challenge conventional peak-finding algorithms. This breakthrough establishes a new paradigm for high-density, distributed FBG sensing networks applicable to land monitoring, soil stability assessment, groundwater detection, maritime surveillance, and smart agriculture.

## 1. Introduction

Optical fiber sensing technologies play a critical role in diverse monitoring applications, ranging from land and vegetation monitoring for disaster response to urban infrastructure surveillance, agricultural biomass assessment, and carbon stock evaluation in ecological systems. These technologies have gained widespread adoption due to their inherent advantages including immunity to electromagnetic interference and capability for long-distance signal transmission [1,2], with particular success demonstrated in geotechnical monitoring, where they outperform conventional sensors in addressing soil material properties and environmental uncertainties [3]. Among these optical sensing technologies, fiber Bragg grating (FBG) sensors have emerged as a particularly robust and versatile solution, widely deployed in structural health monitoring, temperature and strain measurement, and various environmental monitoring applications. The popularity of FBG sensors stems from their comprehensive advantages, including high sensitivity, excellent long-term stability, and inherent wavelength-encoded measurement principle [4,5], with demonstrated superiority over alternative sensing technologies in terms of multiplexing capabilities and reliability in harsh environmental conditions [6,7]. Recent developments have focused on integrating advanced signal processing techniques with FBG sensing systems, including ensemble deep learning approaches and machine learning algorithms for enhanced measurement accuracy [8,9]. Advanced spectral analysis techniques have demonstrated significant improvements in resolving overlapping optical signals through machine learning-enhanced wavelength detection and deconvolution algorithms [10]. Deep learning architectures have shown particular promise in optical sensing applications, with convolutional neural networks achieving superior performance in spectral peak identification [11] and attention-based models excelling in complex spectral feature extraction [12]. These sensors operate by reflecting specific wavelengths of light due to periodic variations in the refractive index within an optical fiber. As external conditions change, the Bragg wavelength shifts, allowing precise tracking of physical parameters. Multiple FBGs with distinct Bragg wavelengths can be inscribed along a single fiber using Wavelength Division Multiplexing (WDM), a technique that enables multiple optical signals at different wavelengths to be transmitted simultaneously over a single optical fiber, thereby forming a distributed sensor network [13,14,15]. However, as the number of sensors increases, spectral crowding becomes a significant issue. Each FBG reflects light over a finite spectral range, defined by its full width at half maximum (FWHM), which can lead to overlap between adjacent Bragg peaks.

To overcome spectral crowding and enhance sensing capacity, some studies have explored Intensity and Wavelength Division Multiplexing (IWDM), which tolerates intentional spectral overlap by distinguishing sensors based on intensity variations. However, the effectiveness of IWDM depends on precise intensity calibration, making it highly susceptible to power fluctuations and environmental noise [16,17]. Additionally, traditional FBG sensing systems typically utilize only the reflection spectrum, while discarding the transmission spectrum, which limits the available information content and sensing capacity. In this study, we propose a hybrid strategy that significantly increases capacity and reliability by integrating both reflection and transmission spectra from FBGs and deploying a dual-linewidth sensor design. While previous work [18] demonstrated that leveraging both reflective and transmission FBG readouts can nearly double sensor capacity within a fixed spectral window, our approach extends this concept by combining it with mixed-linewidth gratings to further enhance multiplexing density. By analyzing both spectra, we double the usable signal content. Furthermore, the system enables inherent self-healing capabilities, where normal transmission mode operation can switch to reflection mode using a circulator when fiber breaks occur, maintaining continuous sensing operation without requiring complex optical switching systems. This differs from existing self-healing approaches [19] that rely on ring architectures with free-space optics, as our method provides fault tolerance through simple mode switching within the same fiber infrastructure.

Additionally, by mixing narrow and broad linewidth FBGs within the same fiber, we add more sensing channels while reducing fabrication costs. Although broad linewidths increase spectral overlap, this is mitigated by our advanced demodulation approach. To reliably decode overlapping and mixed-linewidth spectra, we employ a deep learning demodulation model based on the Transformer architecture [20]. The Transformer uses a self-attention mechanism to extract meaningful features from the spectral data, treating it as a sequence [21,22,23]. This architecture is particularly effective for resolving dense and overlapping spectra across both reflection and transmission modes. The model is trained on a comprehensive simulated dataset that closely replicates real FBG spectral characteristics while simulating bidirectional Bragg wavelength shifts, improving its robustness to both tensile and compressive strain or thermal fluctuations. The Transformer enables accurate and generalizable demodulation across a wide range of sensing conditions. We further enhance system performance by integrating cloud computing into the sensing framework. This allows centralized processing of large volumes of spectral data, supports near-real-time Transformer inference, enables remote access and control, and facilitates integration with other IoT systems [24]. For large-scale or remote deployments, such as pipelines, environmental sensor networks, land monitoring, urban infrastructure surveillance, and agricultural monitoring systems, cloud integration eliminates the need for local computation and provides scalability through on-demand resources. The platform also supports long-term data storage and fusion with other sensing modalities [25].

Figure 1 showcases the core novelty of this work: integrating narrow-band (0.2 nm) and wide-band (0.8 nm) FBGs within the fiber network to boost sensing performance. In Scenario 1, the dual-linewidth array is deployed for comprehensive monitoring: narrow-band FBG sensors (0.2 nm) are bonded to oil storage structures for integrated monitoring of the storage system and the environmental ground conditions underneath, while wide-band FBG sensors (0.8 nm) monitor oil extraction plant infrastructure, with this scenario operating in reflection mode. In Scenario 2, the same dual-linewidth array is deployed for maritime and offshore applications: narrow-band FBG sensors (0.2 nm) monitor the structural integrity and acoustic conditions of oil transportation vessels, while wide-band sensors (0.8 nm) provide critical monitoring of oil refinery platform integrity, with this scenario operating in transmission mode. The specific assignment of narrow-band and wide-band FBGs to different monitoring tasks is based on their complementary characteristics detailed in Section 2. In both cases, the highly overlapped spectra are streamed to a cloud platform, where a Transformer-based demodulation algorithm separates the signals and delivers strain data to a central monitoring office.

The main contributions of this paper are summarized as follows:High-density dual-linewidth FBG network architecture: We propose and validate a sensing framework that multiplies sensor capacity fourfold by combining reflection and transmission analysis with mixed-linewidth FBGs. This method supports ultra-dense, low-cost optical sensing with inherent self-healing capabilities where the system operates normally in transmission mode but changes to reflection mode when fiber breaks occur.Transformer-based demodulation algorithm: We design a deep learning demodulation approach using a Transformer model trained on simulated bidirectional drift scenarios, where drift refers to Bragg wavelength shifts caused by external environmental inputs such as strain, temperature, or pressure variations across both spectral domains. The model achieves precise recovery of overlapping spectral features.Cloud-integrated sensing and analytics: We implement a scalable cloud-based architecture for spectral analysis, centralized data management, and remote sensor network monitoring. This enhances practicality and flexibility for large-scale deployments.

The remainder of this article is organized as follows: Section 2 analyzes the spectral characteristics of our dual-linewidth setup and details synthetic dataset generation. Section 3 presents the system implementation, including cloud integration and the Transformer training pipeline. Section 4 presents the results and discussion under bidirectional drift conditions. Section 5 concludes with a summary and future directions for high-density FBG sensing.

## 2. Linewidth Analysis and Data Collection

Spectral linewidth characteristics fundamentally determine FBG sensor performance, defined as the FWHM of the reflection spectrum. This parameter controls the spectral bandwidth occupied by each sensor, directly affecting multiplexing density, wavelength resolution, and demodulation requirements.

Recent studies have demonstrated that narrow-band gratings concentrate their reflection within tight spectral ranges, enabling precise wavelength tracking and dense sensor packing for high-resolution applications [26,27,28]. In contrast, broader-linewidth gratings occupy wider spectral bands but offer manufacturing advantages and operational robustness through relaxed fabrication tolerances, as shown in fabrication studies [29]. Understanding these characteristics is crucial for designing dual-linewidth sensing systems that leverage the complementary strengths of both approaches. Figure 2 illustrates three representative FBG sets with varying FWHM values. Figure 2a demonstrates narrow-band gratings (0.2 nm) in structural health monitoring and land monitoring applications. Figure 2b shows medium-band gratings (0.5 nm) for distributed sensing in agricultural and urban infrastructure systems. Figure 2c illustrates wide-band gratings (0.8 nm) deployed in harsh maritime environments where manufacturing tolerances and operational stress resistance are critical.

These configurations reveal a fundamental design trade-off: tighter spectral confinement enables higher sensor density but increases sensitivity, whereas broader spectral distribution reduces packing density but improves robustness and simplifies manufacturing. Fabrication constraints reinforce this fundamental trade-off while revealing opportunities for innovative system design. As demonstrated in fabrication research [30,31,32], narrow-linewidth FBGs require longer grating structures and precision inscription techniques like apodization, resulting in higher fabrication costs but superior wavelength resolution, while broad-linewidth gratings utilize shorter structures and standard phase masks, enabling cost-effective mass production with enhanced environmental robustness. Rather than viewing these characteristics as competing limitations, recent demodulation studies have shown that modern approaches can exploit the complementary strengths of both linewidth categories [33,34]. While traditional peak-tracking methods struggle with broad-linewidth spectral overlap, advanced machine learning algorithms can effectively extract meaningful information from mixed-linewidth systems. This paradigm shift transforms the perceived disadvantages of broad FBGs’ wider spectral footprints and tendency to overlap into system advantages when combined with intelligent demodulation, enabling dense sensor arrays that balance cost, robustness, and performance [35,36].

Interrogation methodology significantly influences linewidth selection and system architecture. Traditional approaches pair narrow gratings with high-resolution spectrometers for precise wavelength tracking, while broad gratings typically utilize simplified intensity-based schemes that prioritize speed and cost over resolution. However, these conventional paradigms are being transformed by advanced signal processing techniques. Modern machine learning demodulation can simultaneously handle mixed-linewidth configurations, extracting high-quality information from both narrow and broad spectral features within the same system [37,38]. This capability enables dual-linewidth architectures that combine the precision advantages of narrow gratings with the robustness and cost benefits of broad gratings, transcending traditional linewidth selection constraints to achieve enhanced system capacity and operational flexibility. To simulate the sensing environment, we generate synthetic spectral data using Gaussian models to represent the reflection and transmission characteristics of FBGs. The reflection spectrum of each grating is defined as follows [16]:(1)$$R\left(\lambda ,\lambda_{B,j},\Delta \lambda_{B}\right)=I_{P}\cdot \exp\left[-4\ln2\cdot {\left(\frac{\lambda -\lambda_{B,j}}{\Delta \lambda_{B}}\right)}^{2}\right]$$
where $R\left(\lambda \right)$ is the reflected intensity at wavelength λ; $\lambda_{b,j}$ is the Bragg center wavelength of the j-th FBG; $\Delta \lambda_{B}$ is the full width at half maximum (FWHM); and $R_{\mathrm{peak}}$ is the peak reflectivity of the grating.

For transmission-mode sensing, the spectrum is modeled as the complement of the reflection under an ideal lossless assumption, and is expressed as:(2)$$T\left(\lambda \right)=I_{\mathrm{in}}-R\left(\lambda \right)$$
where $T\left(\lambda ,\lambda_{b,j}\right)$ is the transmitted intensity and $I_{\mathrm{in}}$ is the incident input intensity. This results in a Gaussian-shaped notch centered at the Bragg wavelength. This simplified relation assumes negligible insertion loss and no inter-grating interference. In practical multi-FBG cascaded systems, the transmission spectrum may deviate due to accumulated losses and coupling effects. These equations ensure precise modeling of spectral behavior under varying FWHM and drift conditions.

Table 1 summarizes the spectral and dynamic properties of each grating in Case 1. The parameters include the central Bragg wavelength (λ**_0_**), the FWHM in nm, the peak reflectivity, and the drift role. FBG01 and FBG03 serve as dynamically shifting sensors under positive and negative drift, respectively, while FBG02 acts as a fixed reference. Figure 3 captures the spectral evolution of a uniform-linewidth (0.2 nm) three-FBG array under symmetrical bidirectional drift. FBG01 and FBG03 are dynamically shifted in opposite directions while FBG02 remains fixed. Figure 3a,b show reflection and transmission at ±1 pm drift, respectively, revealing minor peak separations. Figure 3c,d display greater overlap at ±5 pm drift, and Figure 3e,f illustrate increased spectral convergence at ±10 pm drift. The color gradient from purple to yellow visually traces the progressive wavelength shifts, demonstrating the challenges in distinguishing overlapping signals using conventional peak-tracking algorithms. Each subfigure also presents a zoomed-in spectral window (0.10 nm range) to clearly highlight the stepwise wavelength shifts under each drift condition.

Figure 4 presents the spectral responses of a mixed-linewidth (0.2 nm and 0.8 nm) six-FBG array under symmetrical bidirectional drift. FBG11 (narrow) and FBG16 (broad) are shifted, while FBG12–15 remain stationary. Figure 4a,b display moderate spectral separation at ±1 pm drift, with clear peaks in both reflection and transmission. Figure 4c,d show partial merging at ±5 pm drift, as FBG11 begins to overlap with nearby narrow-band gratings and the broad shoulder of FBG16 encroaches on the fixed 0.8 nm group. Figure 4e,f capture the final convergence at ±10 pm drift, where all six peaks blend into a dominant composite band. Each trace corresponds to an incremental drift step, with the color gradient showing the progression from deep purple to bright yellow. A zoomed-in view is included in each subfigure to better visualize how the overall spectral response evolves with each drift condition, particularly highlighting the spectral density, overlap, and shape deformation of peaks as multiple incremental drift steps accumulate. All zoomed-in sections display a consistent 0.05 nm range. These insets help illustrate the transition from distinct, well-separated peaks at low drift to blended or merged profiles at higher drift levels, providing a clearer comparison of spectral coalescence across ±1 pm, ±5 pm, and ±10 pm scenarios. Table 2 provides the full spectral parameters for this configuration. In this setup, FBG11 and FBG16 are the active elements undergoing drift-induced spectral shifts, while the other four gratings—two of each linewidth class—are held static to observe differential overlapping and bandwidth-induced distortion.

Case 1 and Case 2 demonstrate complementary approaches to FBG array design, each addressing different aspects of sensing requirements. Case 1 provides high-resolution tracking with uniform narrow gratings but faces sensor density limitations, while Case 2 shows how mixed linewidths can expand sensing capabilities by leveraging both precision and robustness characteristics. The combination of dual-linewidth arrays creates multiple independent information channels, effectively doubling the sensing capacity within each operational mode by utilizing two linewidth types. This approach addresses the fundamental challenge of balancing sensor density, cost-effectiveness, and measurement precision in large-scale monitoring applications such as land monitoring and maritime infrastructure surveillance. The spectral complexity introduced by these configurations necessitates advanced demodulation techniques capable of resolving overlapping signals and extracting meaningful information from mixed-linewidth systems. To address this challenge, we employ a Transformer-based machine learning algorithm, which will be detailed in the following section, to effectively decode these complex spectral patterns and enable robust peak identification across both narrow and broad bandwidth configurations. Unlike previous studies that focus on single-linewidth optimization or traditional demodulation methods, our novel contribution lies in the systematic integration of dual-linewidth sensing with bidirectional drift simulation across both reflection and transmission spectra, creating a comprehensive dataset specifically designed for Transformer-based demodulation of mixed-linewidth FBG arrays.

## 3. Cloud Computing and Transformer Algorithm

The spectral complexity introduced by dual-linewidth FBG arrays, as demonstrated in the previous section’s case studies, necessitates sophisticated demodulation techniques capable of resolving overlapping signals across both reflection and transmission domains. Traditional peak-tracking algorithms struggle with the dense, overlapping spectral features characteristic of mixed-linewidth systems, particularly under bidirectional drift conditions [39]. To address these challenges, we employ advanced machine learning approaches that can extract meaningful information from complex spectral patterns. The computational demands of training and deploying such algorithms for large-scale monitoring applications spanning land monitoring networks, maritime infrastructure surveillance, and agricultural systems require robust, scalable computing infrastructure. Cloud computing provides an ideal platform for these computationally intensive machines learning tasks, offering dynamic resource allocation, fault tolerance, and centralized processing capabilities essential for real-world FBG sensor deployments [40].

One of its key benefits lies in scalability: cloud infrastructures can dynamically allocate resources such as Graphics Processing Units (GPUs) and memory based on the model’s current workload, avoiding overprovisioning and hardware idle time [41]. This enables cost-effective experimentation with large neural networks [42]. Additionally, cloud environments provide redundancy and fault tolerance, minimizing downtime and ensuring model training continuity in the face of potential hardware failures [43]. Centralized access to storage, containerized environments, and secure collaboration features further enhance reproducibility, development efficiency, and long-term data accessibility, making cloud computing an indispensable tool for modern AI research [44,45]. For this study, all Transformer model training and inference were conducted on Microsoft Azure using a dedicated GPU compute instance. The computational experiments were conducted on a high-performance cloud environment with the following specifications: a Standard_NC24s_v3 virtual machine equipped with 24 cores, 448 GB of RAM, and 2948 GB of SSD storage. The system was powered by four NVIDIA Tesla V100-PCIE-16GB GPUs, running on Ubuntu 22.04.5 LTS, with CUDA 12.2 and driver version 535.247.01.

Model development and experimentation were performed using Azure Machine Learning Studio, with a Python 3.10 kernel and the following core libraries: TensorFlow 2.19.0, Keras 3.9.2, scikit-learn 1.7.0, NumPy 2.1.3, and Pandas 2.2.3. The instance operated under an on-demand pricing model at an approximate cost of $18.97 per hour. To facilitate model development and experimentation, cloud-based tools such as JupyterLab 3.6.8 and Visual Studio Code 1.87.2 (via code-server 1.103.2) were enabled on the instance. 

The instance was configured to provide root access for full software control and was used to train all twelve drift scenarios described in the following sections. The high-end specifications ensured that the Transformer model could process large batches of synthetic spectra efficiently, with low inference delay and high throughput. This scalable architecture is particularly valuable for large-scale deployments across diverse monitoring environments: land monitoring networks with hundreds of distributed sensors, maritime infrastructure requiring real-time structural assessment, and agricultural installations demanding continuous environmental surveillance. The cloud-based approach enables centralized processing of spectral data from geographically dispersed sensor arrays while providing remote access for monitoring personnel.

The Transformer algorithm was selected for its robust performance in sequence transduction tasks and its scalability across large datasets. Unlike traditional recurrent or convolutional networks, the Transformer relies solely on self-attention mechanisms to model long-range dependencies and parallelize training. This architecture makes it ideal for resolving overlapping spectral features in FBG data, where conventional algorithms often struggle. The Transformer architecture is particularly well-suited for FBG spectral demodulation due to several key characteristics. Unlike traditional algorithms that process spectral data sequentially, the self-attention mechanism can simultaneously analyze relationships between all wavelength points, enabling effective resolution of overlapping peaks from multiple gratings. This global receptive field is crucial for mixed-linewidth systems where narrow and broad spectral features interact across the entire wavelength range. Additionally, the multi-head attention mechanism allows the model to focus on different spectral characteristics simultaneously, tracking sharp narrow-band peaks while monitoring broad-band envelope changes, making it highly effective for dual-linewidth configurations in diverse monitoring applications. Furthermore, the Transformer’s versatility in processing both reflection and transmission spectra makes it well-suited for systems with self-healing capabilities. When fiber breaks occur and the system changes from normal transmission mode operation to reflection mode through self-healing mechanisms, the Transformer can seamlessly handle the spectral data from either operational mode, ensuring continuous monitoring capability.

The Transformer model consists of several key components:

**Positional Encoding**: Because the Transformer lacks recurrence or convolution, it cannot inherently understand the order of the sequence. To address this, sinusoidal positional encodings are added to the input embeddings [20]:(3)$$PE_{\left(pos,2i\right)}=sin\left(\frac{pos}{10000^{2i/d_{model}}}\right)\;PE_{\left(pos,2i+1\right)}=cos\left(\frac{pos}{10000^{2i/d_{model}}}\right)$$
where pos is the position index in the sequence; i is the dimension index of the embedding; and $d_{model}$ is the total embedding dimension. These encodings inject position-dependent variations that help the model distinguish between sequence elements and capture sequential structure.

**Scaled Dot–Product Attention**: This mechanism computes a weighted representation of the input by comparing queries (Q) to keys (K) and applying the result to values (V) [20]:(4)$$Attention\left(Q,K,V\right)=\mathrm{softmax}\left(\frac{QK^{T}}{\sqrt{d_{k}}}\right)V$$
where Q is the matrix of queries; K is the matrix of keys; V is the matrix of values; and $d_{k}$ is the dimensionality of the key vectors. The dot product, $QK^{T},$ quantifies similarity, $\sqrt{d_{k}}$ is a scaling factor to prevent large gradient variance, and softmax ensures proper normalization into attention weights.

**Multi-Head Attention**: Instead of relying on a single attention mechanism, the model uses multiple parallel attention heads [20]:(5)$$\mathrm{MultiHead}\left(Q,K,V\right)=\mathrm{Concat}\left(head_{1},\ldots ,head_{h}\right)W^{O}$$
where(6)$$head_{i}=Attention\left(QW_{i}^{Q},KW_{i}^{K},VW_{i}^{V}\right)$$
where $W_{i}^{Q},W_{i}^{K},W_{i}^{V}$ are the learned linear projection matrices for the ith head; $W^{O}$ is the learned output projection matrix; and h is the number of attention heads. This design allows the model to attend to information from different subspaces simultaneously, capturing diverse relationships within the input.

**Feed-Forward Networks (FFN)**: Each position in the sequence is independently passed through a fully connected feed-forward network [20]:(7)$$FFN\left(x\right)=max\left(0,xW_{1}+b_{1}\right)W_{2}+b_{2}$$
where x is the input vector at a specific position; $W_{1},W_{2}$ are the weight matrices; $b_{1},b_{2}$ are the bias vectors; and $max\left(0,\cdot \right)$ is the ReLU activation. This adds non-linearity and depth to the model, enabling it to learn complex mappings beyond simple attention aggregation.

**Add and Normalize**: To stabilize training, each sub-layer uses residual connections followed by layer normalization [20]:(8)$$\mathrm{LayerNorm}\left(x+\mathrm{Sublayer}\left(x\right)\right)$$
where x is the input to the sub-layer; $\mathrm{Sublayer}\left(x\right)$ is the output of the attention or feed-forward module; and LayerNorm is the layer normalization function. This ensures efficient gradient flow and numerical stability across stacked layers.

The Transformer model architecture incorporates a positional encoding function to inject sequential order into the input data. The model architecture incorporates sinusoidal positional encoding, which assigns unique position-based vectors to each input token by applying sine functions to even indices and cosine functions to odd indices across the Transformer architecture, which utilizes multi-head self-attention, residual connections, layer normalization, and a two-layer feed-forward network.

In decoder mode, it additionally integrates cross-attention mechanisms over the encoder outputs. Key hyperparameters for training include a learning rate of 0.0001, batch size of 36, dropout rate of 0.5, and an L2 regularization factor of 0.005. The model is configured with two attention heads, a feed-forward network dimension of 102, and an embedding size of 22. Training is carried out over 1000 epochs with gradient clipping set to 1.0, using the Adam optimizer. The architecture consists of a single Transformer block each for the encoder and decoder. The encoder normalizes the input data, while the decoder integrates both self-attention and context from the encoder. The final dense layers compress the output to three values. Compilation is performed with the Adam optimizer (with gradient clipping), which uses the mean squared error as the loss function and performance metrics including mean absolute error and accuracy. GPU logs confirm the successful allocation of four NVIDIA Tesla V100 GPUs for accelerated training. Key strengths of the Transformer include its parallelism, which drastically reduces training time; its global receptive field, enabling better resolution of closely spaced peaks; and its generalization capability, which is critical when predicting under varying FWHM and spectral drift conditions. In summary, the integration of Azure-based cloud infrastructure with Transformer modeling provides a scalable and efficient platform for high-fidelity FBG signal demodulation. The next section details the results of this implementation across Cases 1 and 2.

## 4. Results and Discussion

Continuing from the end of Section 3, this section presents the comprehensive evaluation of the Transformer-based demodulation system across both uniform Case 1 and mixed-linewidth Case 2 FBG arrays under various spectral drift conditions. The results demonstrate the effectiveness of the proposed approach in handling complex overlapping scenarios that traditional peak-finding algorithms struggle to resolve. Figure 5 illustrates the robust training characteristics of the Transformer demodulator when applied to the six drift scenarios of Case 2, which represents the more challenging mixed-linewidth configuration. Figure 5a shows the training and validation loss curves, demonstrating excellent convergence behavior, with final loss values as low as 0.00022 for the most stable scenario. Figure 5b shows the model achieves remarkable demodulation accuracy, with five out of six scenarios reaching perfect accuracy (1.0) and the remaining scenario achieving 99.8% accuracy.

This exceptional performance validates the Transformer’s ability to learn complex spectral patterns and generalize effectively across different drift magnitudes. The consistent convergence across all scenarios, despite varying degrees of spectral overlap, highlights the architecture’s inherent stability and robustness in handling the non-linear relationships between overlapped FBG spectra and their constituent peak positions. Figure 6 provides crucial insights into the Transformer’s performance across all twelve drift scenarios encompassing both Cases 1 and 2. Figure 6a reveals distinct performance characteristics between the two cases and operational modes through Mean Squared Error (MSE) analysis. Case 1 demonstrates superior accuracy with MSE values ranging from 0.00268 to 0.00997, while Case 2 shows slightly higher but still acceptable MSE values between 0.01277 and 0.02534. This difference is attributed to the increased complexity introduced by the mixed-linewidth configuration in Case 2, where the varying spectral characteristics of FBG11 and FBG16 create more challenging demodulation conditions.

The transmission mode generally exhibits comparable or slightly higher MSE values compared to reflection mode, indicating that while both modes are viable, reflection mode maintains a marginal accuracy advantage. Figure 6b shows the Root Mean Squared Error (RMSE) metrics further corroborate these findings, with Case 1 achieving RMSE values between 0.04971 and 0.05971, while Case 2 ranges from 0.06171 to 0.08917. These values represent excellent accuracy considering the challenging nature of overlapped spectral demodulation. Figure 6c presents the Mean Absolute Error (MAE) analysis, showing consistent trends, with Case 1 maintaining MAE values below 0.10171 and Case 2 reaching maximum values of 0.15917. Figure 6d demonstrates that the Mean Percentage Error (MPE) values remain remarkably low across all scenarios, with magnitudes typically below 0.01, demonstrating the absence of systematic bias in the Transformer’s predictions. The near-zero MPE values, including both positive and negative deviations, confirm that the model provides unbiased estimates without consistent over- or under-prediction tendencies. Figure 7 provides comprehensive insights into the computational requirements and scalability characteristics of the Transformer-based approach. Figure 7a reveals an interesting pattern where initial scenarios at 1 pm require significantly longer training periods (approximately 3890–3899 ms) compared to subsequent time points. The 5 pm scenarios demonstrate reduced training times (906–913 ms), while the 10 pm scenarios achieve the fastest convergence (530–537 ms). This temporal efficiency improvement suggests that the model benefits from transfer learning effects and optimized weight initialization as training progresses through the sequential drift scenarios. Figure 7b shows the inference testing times remain consistently efficient across all scenarios, ranging from 1.13768 to 1.21718 s, demonstrating the real-time capability of the system.

The remarkable consistency in testing times, regardless of spectral complexity or drift magnitude, highlights the Transformer’s parallel processing advantages and suggests its potential suitability for near-real-time sensing, provided that network and system latencies are appropriately managed. Figure 7c demonstrates that the mean-training-time aggregation shows minimal difference between Case 1 and Case 2, indicating that the mixed-linewidth configuration does not significantly impact computational requirements despite its increased spectral complexity. Figure 7d presents the dataset size analysis, revealing an adaptive training strategy where 1801 spectra are utilized for the most challenging 1 pm scenarios, 361 spectra for intermediate 5 pm conditions, and 181 spectra for the 10 pm scenarios. This graduated approach optimizes training efficiency while maintaining robust performance across all drift conditions, demonstrating the system’s ability to adapt computational resources based on problem complexity.

Figure 8 showcases the practical effectiveness of the Transformer demodulator through visual representation of predicted peak wavelengths under symmetrical bidirectional drift conditions. Figure 8a,b demonstrate that the reflection and transmission spectra at ±1 pm drift show clear peak identification capabilities, with the model successfully distinguishing between FBG11 (+Δλ) and FBG16 (−Δλ) despite their proximity. Figure 8c,d present the results as the drift increases to ±5 pm; the spectra exhibit partial overlap conditions that would challenge conventional peak-finding algorithms, yet the Transformer maintains accurate peak localization in both reflection and transmission modes. Figure 8e,f show the most impressive demonstration occurs at ±10 pm drift, where the gratings experience significant spectral overlap. In reflection mode, the Transformer successfully identifies isolated peaks from what appears as a single composite spectral feature to traditional analysis methods. The corresponding transmission spectra show successful peak identification within broad composite pass-bands, confirming the model’s ability to decompose complex overlapped signals into their constituent components.

This capability represents a significant advancement over conventional FBG demodulation techniques, which typically fail under such severe overlap conditions. The consistent performance across both reflection and transmission modes validates the versatility of the approach and opens new possibilities for sensor array design. The ability to accurately demodulate signals in transmission mode, traditionally considered less favorable due to noise considerations, expands the operational flexibility of FBG sensing systems. From these comprehensive results, we can confidently conclude that the Transformer-based approach effectively handles dual-linewidth systems while delivering superior accuracy compared to traditional methods. The quantitative performance comparison reveals the Transformer architecture’s distinct advantage, achieving average MSE values of 0.003793 and 0.008903 for Case 1 reflection and transmission modes, respectively, and 0.018683 and 0.016747 for Case 2 configurations. In contrast, LSTM networks demonstrated considerably higher error rates with MSE values of 0.019793 and 0.028903 for Case 1, and 0.08683 and 0.096747 for Case 2. Similarly, GRU networks, while performing better than LSTM, still exhibited elevated MSE values of 0.009793 and 0.018903 for Case 1, and 0.06683 and 0.056747 for Case 2. This consistent pattern of lower error rates across all operational scenarios underscores the Transformer’s enhanced capability for processing complex spectral overlaps in FBG sensor arrays. The successful integration of mixed linewidth configurations demonstrates the system’s adaptability to diverse FBG specifications, making it suitable for complex sensor networks with heterogeneous components. This work demonstrates that the dual-linewidth approach effectively doubles the sensing capacity within each operational mode (reflection mode or transmission mode) by utilizing both narrow (0.2 nm) and broad (0.8 nm) linewidth gratings. Combining both reflection and transmission modes simultaneously can theoretically quadruple sensor capacity, effectively achieving a potential increase where dual-linewidth benefits are ideally multiplied by dual-mode operation. However, practical constraints such as spectral overlap and system noise may limit the achievable gain in real-world deployments. This dual-linewidth approach represents a significant advancement in FBG sensor array design, enabling enhanced sensor density while maintaining individual sensor resolution and accuracy.

## 5. Conclusions

This work demonstrates the effectiveness of Transformer-based neural networks for dual-linewidth FBG sensor demodulation, achieving exceptional accuracy in resolving overlapping spectral signatures across bidirectional drift conditions. The integration of reflection and transmission mode analysis with mixed-linewidth configurations successfully doubles sensor capacity while enabling inherent self-healing capabilities through operational mode changes during fiber breaks. The cloud-based implementation provides scalable processing for large-scale monitoring networks spanning land monitoring, maritime infrastructure surveillance, and agricultural installations. The demonstrated robustness under severe spectral overlap conditions and consistent computational efficiency establish this approach as a practical solution for dense FBG sensor deployments. Future work should explore extended wavelength ranges and integration with additional sensing modalities to further enhance system capabilities.

## Figures and Tables

**Figure 1 sensors-25-05627-f001:**
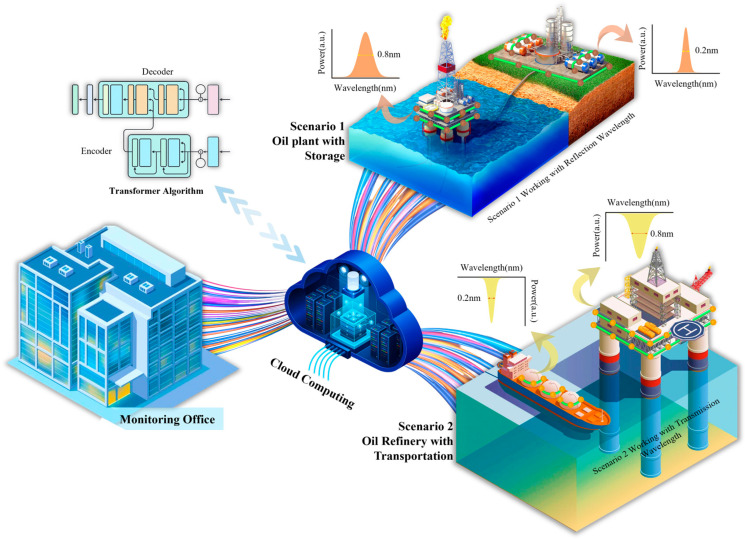
Conceptual diagram of the proposed dual-linewidth FBG sensing network.

**Figure 2 sensors-25-05627-f002:**
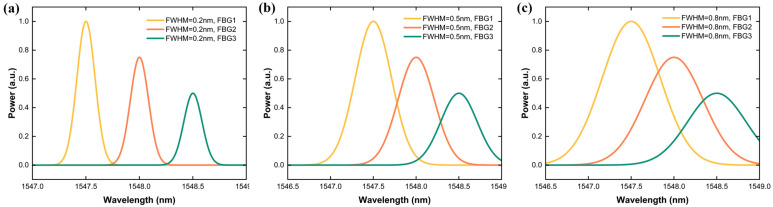
Reflection spectra of three FBG sets with different linewidths: (**a**) narrow-band gratings with 0.2 nm FWHM; (**b**) medium-band gratings with 0.5 nm FWHM; (**c**) wide-band gratings with 0.8 nm FWHM.

**Figure 3 sensors-25-05627-f003:**
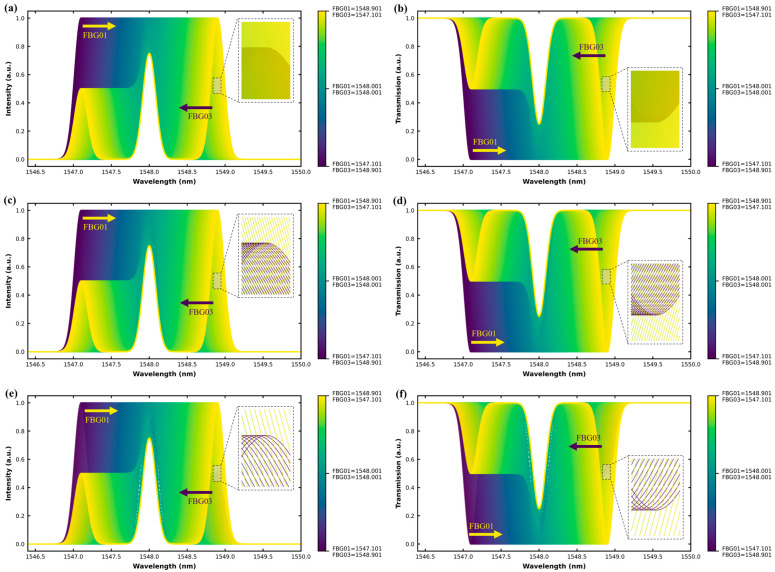
Spectral responses of the uniform-linewidth three-FBG array (Case 1) under symmetrical bidirectional drift applied to FBG01 and FBG03, with FBG02 fixed: (**a**) reflection spectrum at ±1 pm drift; (**b**) transmission spectrum at ±1 pm drift; (**c**) reflection spectrum at ±5 pm drift; (**d**) transmission spectrum at ±5 pm drift; (**e**) reflection spectrum at ±10 pm drift; (**f**) transmission spectrum at ±10 pm drift. Each trace represents a specific incremental drift step, and the color gradient visualizes the stepwise spectral changes from deep purple to bright yellow. A zoomed-in view is included in each subfigure to better visualize the overall stepwise spectral shifts under each drift condition.

**Figure 4 sensors-25-05627-f004:**
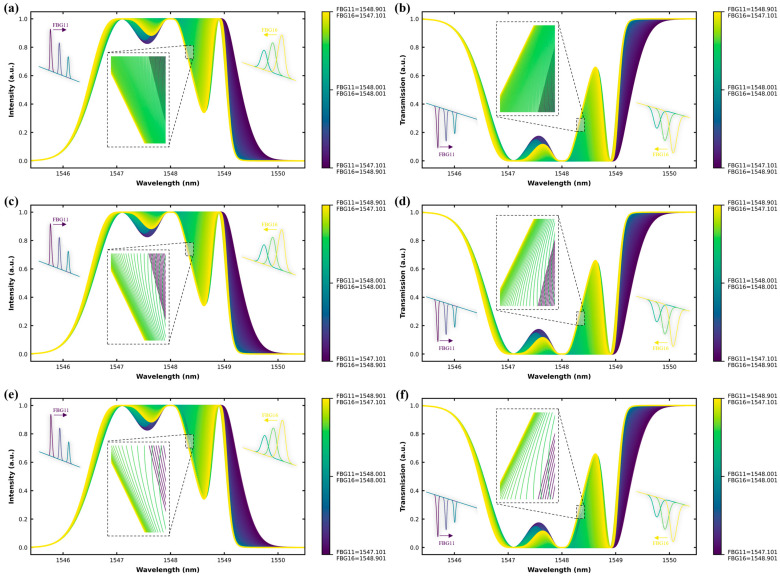
Spectral responses of the mixed-linewidth six-FBG array (Case 2) under symmetrical bidirectional drift applied to FBG11 and FBG16, with FBG12–FBG15 fixed: (**a**) reflection spectrum at ±1 pm drift; (**b**) transmission spectrum at ±1 pm drift; (**c**) reflection spectrum at ±5 pm drift; (**d**) transmission spectrum at ±5 pm drift; (**e**) reflection spectrum at ±10 pm drift; (**f**) transmission spectrum at ±10 pm drift. Each trace corresponds to an incremental drift step, with the color gradient showing the progression from deep purple to bright yellow. A zoomed-in view is included in each subfigure to better visualize the overall stepwise spectral shifts under each drift condition.

**Figure 5 sensors-25-05627-f005:**
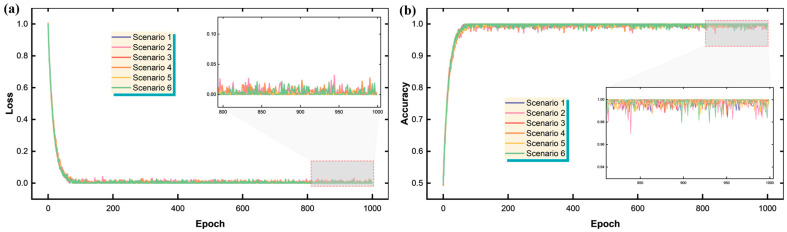
Performance of the Transformer demodulator on the six drift scenarios of Case 2 (mixed-linewidth array): (**a**) training and validation loss versus epoch; (**b**) final demodulation accuracy for the same six scenarios from Case 2.

**Figure 6 sensors-25-05627-f006:**
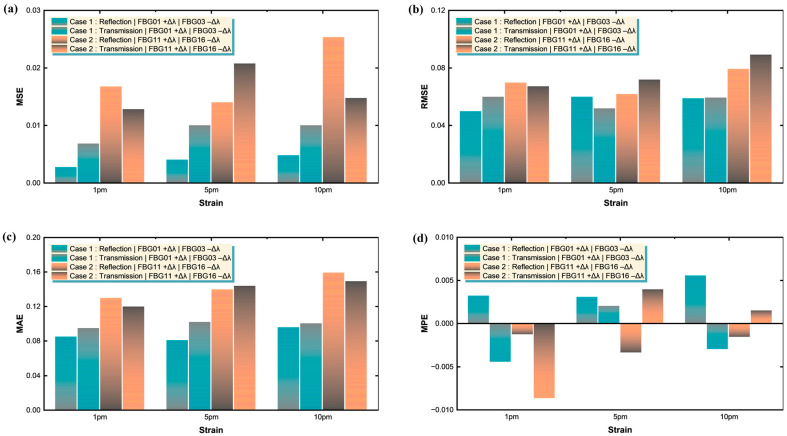
Comprehensive error analysis of the Transformer demodulator across all twelve drift scenarios (Cases 1 & 2): (**a**) Mean squared error (MSE); (**b**) root-mean-squared error (RMSE); (**c**) mean absolute error (MAE); (**d**) mean percentage error (MPE).

**Figure 7 sensors-25-05627-f007:**
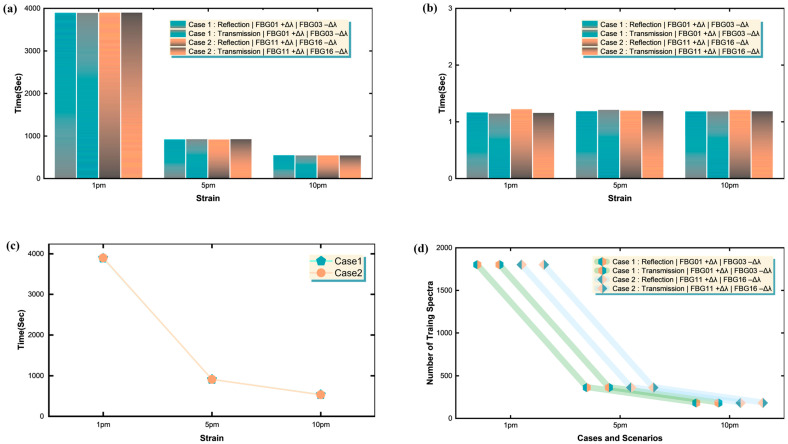
Computational cost and dataset size for Transformer training and inference across the twelve drift scenarios (Cases 1 and 2): (**a**) GPU training time per scenario; (**b**) inference (testing) time per scenario; (**c**) mean training time aggregated by case (uniform versus mixed linewidth array); (**d**) number of synthetic training spectra generated for each scenario.

**Figure 8 sensors-25-05627-f008:**
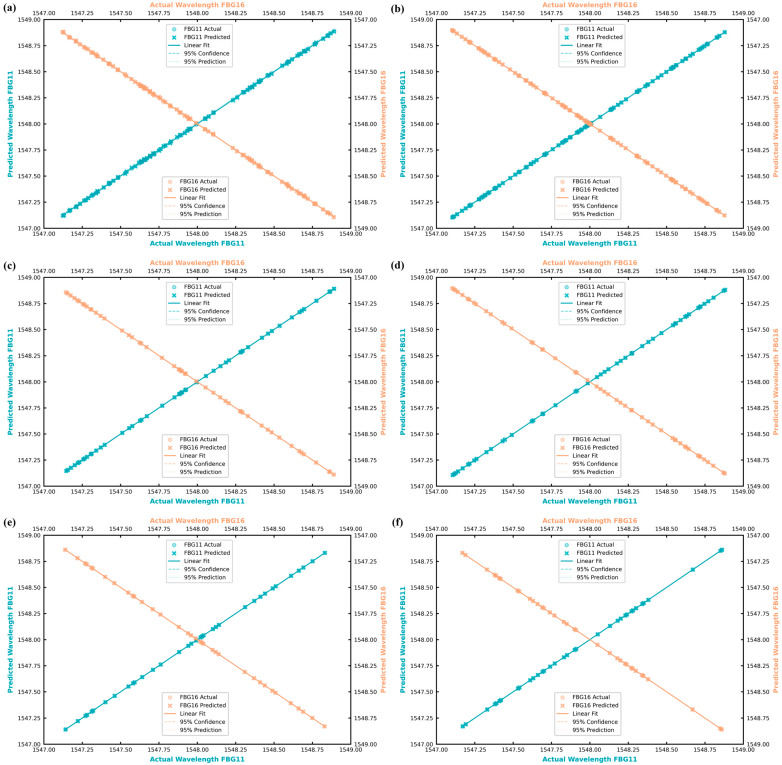
Transformer-predicted peak wavelengths for the strained gratings in Case 2 (mixed-linewidth array) under symmetrical bidirectional drift applied to FBG11 and FBG16: (**a**) reflection spectrum at ±1 pm drift with predicted peaks of FBG11 (+Δλ) and FBG16 (−Δλ); (**b**) transmission spectrum at ±1 pm drift showing corresponding notch detections; (**c**) reflection spectrum at ±5 pm drift with peak localization amid partial overlap; (**d**) transmission spectrum at ±5 pm drift confirming dual-peak detection; (**e**) reflection spectrum at ±10 pm drift with isolated peaks from overlapping gratings; (**f**) transmission spectrum at ±10 pm drift showing successful peak identification within the broad composite pass-band.

**Table 1 sensors-25-05627-t001:** Case 1—Bidirectional drift matrix for a three-FBG array with uniform 0.20 nm linewidth (FWHM).

FBG ID	Centre λ_0_ (nm)	FWHM (nm)	Peak R (a.u.)	Drift Role
FBG01	1547.501	0.20	1.00	drifting (+Δλ)
FBG02	1548.001	0.20	0.75	fixed (Δλ = 0)
FBG03	1548.901	0.20	0.50	drifting (−Δλ)

**Table 2 sensors-25-05627-t002:** Case 2—Bidirectional drift matrix for a mixed-linewidth six-FBG array (0.20 nm and 0.80 nm FWHM).

FBG ID	Centre λ_0_ (nm)	FWHM (nm)	Peak R (a.u.)	Drift Role
FBG11	1547.101	0.2	1.00	drifting (+Δλ)
FBG12	1548.001	0.2	0.75	fixed (Δλ = 0)
FBG13	1548.901	0.2	0.50	fixed (Δλ = 0)
FBG14	1547.101	0.8	1.00	fixed (Δλ = 0)
FBG15	1548.001	0.8	0.75	fixed (Δλ = 0)
FBG16	1548.901	0.8	0.50	drifting (−Δλ)

## Data Availability

The data presented in this study are available in this article.

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
