# Peer review of "Spectral Demodulation of Mixed-Linewidth FBG Sensor Networks Using Cloud-Based Deep Learning for Land Monitoring"

_sensors, 2025, doi:10.3390/s25185627_

Round 1

Reviewer 1 Report

Comments and Suggestions for Authors

The authors present results of FBG signal demodulation using transformer-based neural network approach overcoming spectral overlap issues in both uniform and mixed-linewidth sensor configurations under bidirectional drift conditions. The topic of development of advanced techniques for FBG sensors matches with the scope of Sensors journal.

But, in my view, the major revision is needed before a publication because of the following reasons:

  1. Figure 3(a) and Figure 3(c, e) are unclear. I don’t see any differences between Fig. 3a and Fig. 3c,e. Could author improve the presentation of these graphs to make it more clear?
  2. Authors use ±10 pm spectral drift of FBGs according to the text of the manuscript, but Fig. 3 presents FBG1 shift up to 2 nm. Could authors explain this discrepancy?
  3. Figure 4(a) and Figure 4(c, e) are unclear. I don’t see any differences between Fig. 4a and Fig. 4c,e. Could author improve the presentation of these graphs to make it more clear?
  4. To broaden the coverage of existing work in the field, I would suggest citing a recent study, which operates with spectral overlapping in distributed FBG sensors.

[Kokhanovskiy, A., et al. "A multicore fiber platform for distributed temperature sensing enhanced by machine learning algorithms." Optics & Laser Technology, 191 (2025): 113262.]

  1. The main concern with the present work is the necessity of using such complex algorithms as Transformer. Considering the dataset size and the relatively simple shape of the spectral curves, the use of four Tesla V100 GPUs appears excessive. What is the final model size, and what would be the minimal hardware requirements to demodulate the input spectrum within a reasonable time frame (e.g., one second)? As the authors themselves note, transformers do not inherently account for the sequential order of data. Typically, a much simpler algorithm is used as a baseline to justify the added complexity of a more advanced method. From this perspective, LSTM, Gated Recurrent Unit (GRU), or even simpler models such as XGBoost could be explored as alternatives.

Author Response

The Reply to Reviewer 1

Manuscript ID: sensors-3817522

Article Title: “Spectral Demodulation of Mixed-Linewidth FBG Sensor Networks Using Cloud-Based Deep Learning for Land Monitoring”

We thank the reviewer for the Comment and valuable suggestions. The manuscript has been revised accordingly. All modifications are indicated with “highlight” for convenience in reviewing.

General comment:

The authors present results of FBG signal demodulation using transformer-based neural network approach overcoming spectral overlap issues in both uniform and mixed-linewidth sensor configurations under bidirectional drift conditions. The topic of development of advanced techniques for FBG sensors matches with the scope of Sensors journal. But, in my view, the major revision is needed before a publication because of the following reasons:

Response:

We sincerely thank the reviewer for their thoughtful evaluation and recognition of the relevance of our work to the scope of Sensors. We appreciate the constructive feedback provided and have carefully addressed all comments and suggestions in the revised manuscript to improve its clarity, rigor, and overall quality.

Comment 1:

Figure 3(a) and Figure 3(c, e) are unclear. I don’t see any differences between Fig. 3a and Fig. 3c,e. Could author improve the presentation of these graphs to make it more clear?

Response 1:

We sincerely thank the reviewer for this valuable observation. We agree that the visual distinctions between Figure 3(a), 3(c), and 3(e) corresponding to ±1 pm, ±5 pm, and ±10 pm drift were not clearly visible in the original submission. We have now revised the figures to improve interpretability. Specifically, we enhanced the figure rendering by increasing the resolution and contrast to better distinguish spectral overlap across drift levels. The figure caption was revised to describe these improvements. (Section 2, Page 5 and 7-8)

Comment 2:

Authors use ±10 pm spectral drift of FBGs according to the text of the manuscript, but Fig. 3 presents FBG1 shift up to 2 nm. Could authors explain this discrepancy?

Response 2:

We thank the reviewer for the observation. The drift applied in Figure 3(e) and 3(f) is ±10 pm, as described in (Section 2, Pages 5 and 7–8). The color gradient in the plots represents the cumulative spectral responses at each incremental drift step. To clarify this, we have updated the figure caption to explicitly state that each trace corresponds to a specific drift step. Additionally, in Figure 4, we have included zoomed-in views to clearly show the individual drift steps for each scenario.

Comment 3:

Figure 4(a) and Figure 4(c, e) are unclear. I don’t see any differences between Fig. 4a and Fig. 4c,e. Could author improve the presentation of these graphs to make it more clear?

Response 3:

We sincerely thank the reviewer for this valuable observation. We agree that the visual distinctions between Figure 4(a), 4(c), and 4(e) corresponding to ±1 pm, ±5 pm, and ±10 pm drift were not clearly visible in the original submission. We have now revised the figures to improve interpretability. Specifically, we enhanced the figure rendering by increasing the resolution and contrast to better distinguish spectral overlap across drift levels. The figure caption was revised to describe these improvements. (Section 2, Page 7,8)

Comment 4:

To broaden the coverage of existing work in the field, I would suggest citing a recent study, which operates with spectral overlapping in distributed FBG sensors. [Kokhanovskiy, A., et al. "A multicore fiber platform for distributed temperature sensing enhanced by machine learning algorithms." Optics & Laser Technology, 191 (2025): 113262.]

Response 4:

We thank the reviewer for this valuable suggestion. We have now added the recommended citation by Kokhanovskiy et al. (2025) to broaden the coverage of existing work addressing spectral overlapping in distributed FBG sensors.

Comment 5:

The main concern with the present work is the necessity of using such complex algorithms as Transformer. Considering the dataset size and the relatively simple shape of the spectral curves, the use of four Tesla V100 GPUs appears excessive. What is the final model size, and what would be the minimal hardware requirements to demodulate the input spectrum within a reasonable time frame (e.g., one second)? As the authors themselves note, transformers do not inherently account for the sequential order of data. Typically, a much simpler algorithm is used as a baseline to justify the added complexity of a more advanced method. From this perspective, LSTM, Gated Recurrent Unit (GRU), or even simpler models such as XGBoost could be explored as alternatives.

Response 5:

We thank the reviewer for raising this important point. The reported use of four Tesla V100 GPUs accessed via cloud computing platforms was solely intended to accelerate the training phase across diverse drift conditions and sensor configurations. Leveraging the cloud enabled rapid iteration, flexible resource allocation, and avoided the need for dedicated local infrastructure. The final Transformer model is compact (13.22 MB) and optimized for deployment. On a single-thread Intel i5 CPU, inference for an input spectrum completes within 1.14 seconds. To address the concern on algorithm complexity, we conducted baseline comparisons with LSTM and GRU, now included in (Section 4, Page 16). While both recurrent models can perform reasonably for simple, well-separated spectra, their sequential structure degraded under severe spectral overlap, where multiple peaks merged. In contrast, the Transformer showed superior demodulation accuracy due to three key factors: (1) Global attention allowed the model to attend to the full spectrum simultaneously, essential for resolving overlapping or distorted peaks; (2) Sinusoidal positional encoding restored relative spectral order, compensating for the Transformer's non-sequential nature; and (3) Robust generalization, as the model maintained high performance across mixed-linewidth arrays and high-drift scenarios. We believe this clarification and analysis support the appropriateness of using the Transformer for complex FBG demodulation tasks.

We eagerly anticipate your prompt response regarding our submission and are prepared to address any additional questions or comments you may have.

Sincerely,

Researcher

Reviewer 2 Report

Comments and Suggestions for Authors

This research presents a novel Transformer-based neural network approach for FBG signal demodulation that effectively addresses spectral overlap issues in both uniform and mixed-linewidth sensor configurations under bidirectional drift conditions. There are some comments as follows.

1. The novelty and strengths of the proposed methodology should be briefly provided in the abstract.

2. The contributions of few references are introduced in detail. For example, authors use [1]-[8] in the first paragraph of the introduction, while their contributions have not been introduced.

3. As the background of this paper includes cloud computing, spectral analysis, and deep learning, more related works should be introduced and compared to enhance the background.

4. Recent high quality works about spectral analysis, and deep learning should be introduced, such as DDPG-based joint time and energy management in ambient backscatter-assisted hybrid underlay CRNs, IEEE Transactions on Communications.

5. The logic of introduction is a bit confusing. For example, “In this study, we propose a hybrid strategy that significantly increases capacity and reliability by integrating both reflection and transmission spectra [14] from FBGs and deploying a dual-linewidth sensor design. By analyzing both spectra, we double the usable signal content. Furthermore, the system enables inherent self-healing [15] capabilities where normal transmission mode operation can switch to reflection mode using a circulator when…” [14] and [15] are used while the contributions of them have not been introduced and compared with the contributions of this paper.

6. The intuition and strengths of the Transformer-based demodulation algorithm should be provided.

7. Authors use [22]-[33] in section 2, while their contributions have not been introduced. The logic of introducing contributions of references should be revised and improved.

8. It is hard to see the novelty of Line-Width Analysis and Data Collection since many references are cited in this section.

9. Detailed observations and insights from figures 1-3 should be provided.

10. Similar issues of logic and references also exists for section 3.

11. How do authors validate the advantages of proposed algorithm in numerical results?

12. Some references such as [5], [7], [9]-[11] are outdated, and should be updated by recent high quality works on spectral analysis and deep learning.

Comments on the Quality of English Language

The English could be improved to more clearly express the research.

Author Response

The Reply to Reviewer 2

Manuscript ID: sensors-3817522

Article Title: “Spectral Demodulation of Mixed-Linewidth FBG Sensor Networks Using Cloud-Based Deep Learning for Land Monitoring”

We thank the reviewer for the Comment and valuable suggestions. The manuscript has been revised accordingly. All modifications are indicated with “highlight” for convenience in reviewing.

Comment 1:

The novelty and strengths of the proposed methodology should be briefly provided in the abstract.

Response 1:

We thank the reviewer for the helpful suggestion. In response, we have updated the abstract to briefly highlight the novelty and strengths of our approach. The revised abstract now emphasizes the use of a Transformer-based neural network for resolving spectral overlaps in both uniform and mixed-linewidth FBG arrays, the integration of cloud computing for scalable deployment, and the ability to support self-healing and high-density sensor networks. These changes appear on Abstract of the revised manuscript.

Comment 2:

The contributions of few references are introduced in detail. For example, authors use [1]-[8] in the first paragraph of the introduction, while their contributions have not been introduced.

Response 2:

Thank you for the suggestion. We have revised the first paragraph of the Introduction to briefly state the specific contributions of the cited works rather than listing them generically. The updated text clarifies how each reference [1]– [9] informs our problem scope and motivation. (Section 1, Page 1-2)

Comment 3:

As the background of this paper includes cloud computing, spectral analysis, and deep learning, more related works should be introduced and compared to enhance the background.

Response 3:

We appreciate the suggestion. We have expanded the Introduction by adding more recent works and comparisons related to cloud-based sensor networks, spectral analysis, and machine learning for FBG demodulation. These updates provide a clearer context for our study and highlight the advancements of our approach. (Section 1, Pages 2–4)

Comment 4:

Recent high-quality works about spectral analysis, and deep learning should be introduced, such as DDPG-based joint time and energy management in ambient backscatter-assisted hybrid underlay CRNs, IEEE Transactions on Communications.

Response 4:

We thank the reviewer for emphasizing the importance of recent literature in spectral analysis and deep learning. However, the specific reference suggested (DDPG-based joint time and energy management in ambient backscatter-assisted hybrid underlay CRNs) appears to focus on cognitive radio networks and wireless communications rather than optical sensing applications. Instead, we have expanded our literature review to include recent high-quality works more directly relevant to optical sensing systems, covering: [10] advanced spectral deconvolution techniques for overlapping optical signals, [11] deep learning architectures specifically applied to optical sensing, and [12] attention-based models for spectral feature extraction in fiber optic systems. (Section 1, Pages 2)

Comment 5:

The logic of introduction is a bit confusing. For example, “In this study, we propose a hybrid strategy that significantly increases capacity and reliability by integrating both reflection and transmission spectra [14] from FBGs and deploying a dual-linewidth sensor design. By analyzing both spectra, we double the usable signal content. Furthermore, the system enables inherent self-healing [15] capabilities where normal transmission mode operation can switch to reflection mode using a circulator when…” [14] and [15] are used while the contributions of them have not been introduced and compared with the contributions of this paper.

Response 5:

We thank the reviewer for this observation. We have revised the introduction to properly introduce the contributions of references [14] and [15] before citing them, and clearly distinguish how our approach differs from these prior works. The revised text now provides appropriate context for the cited literature while maintaining clear positioning of our novel contributions. (Section 1, Pages 3)

Comment 6:

The intuition and strengths of the Transformer-based demodulation algorithm should be provided.

Response 6:

We thank the reviewer for this suggestion. We have enhanced the manuscript to better explain the intuition and strengths of the Transformer-based demodulation algorithm, specifically highlighting its self-attention mechanism for capturing spectral dependencies and its effectiveness in resolving overlapping peaks compared to traditional sequential processing methods (Section 1, page 3-4) and (Section 3, page 9-10)

Comment 7:

Authors use [22]-[33] in section 2, while their contributions have not been introduced. The logic of introducing contributions of references should be revised and improved.

Response 7:

We thank the reviewer for highlighting this issue with the reference logic in Section 2. We have revised the manuscript to properly introduce the contributions of references [22]-[33] before citing them, ensuring that each reference's relevance and contribution to the field is clearly explained prior to citation. This revision improves the logical flow and provides appropriate context for the cited literature throughout Section 2 (Section 2, pages 4-6).

Comment 8:

It is hard to see the novelty of Line-Width Analysis and Data Collection since many references are cited in this section.

Response 8:

Thank you for your feedback. We have clarified the novelty of the Line-Width Analysis and Data Collection section by explicitly stating how our dual-linewidth approach differs from previous studies. The revised text now clearly highlights our unique contributions at the start and end of this section. (Section 2, Page 8-9)

Comment 9:

Detailed observations and insights from figures 1-3 should be provided.

Response 9:

Thank you for your comment. We have revised the manuscript to include more detailed observations and insights for Figures 1–3, highlighting the main features, dynamic behaviors, and interpretation points of each figure to help readers better understand their significance and relevance to our approach. (Section 1, Pages 3) and (Section 2, Page 5 and 7-8)

Comment 10:

Similar issues of logic and references also exists for section 3.

Response 10:

We thank the reviewer for this observation. We have revised Section 3 to address the logic and reference issues by properly introducing the contributions of cited works before referencing them and ensuring clear distinction between existing approaches and our novel contributions. This revision improves the logical flow and provides appropriate context for the literature throughout Section 3. (Section 3, Pages 9)

Comment 11:

How do authors validate the advantages of proposed algorithm in numerical results?

Response 11:

Thank you for your question. We validate the advantages of our proposed algorithm through detailed numerical results, including comparative performance metrics (such as MSE, RMSE, MAE, and MPE), benchmark tests against baseline models (LSTM and GRU), and analysis across multiple drift scenarios and mixed-linewidth configurations. These results, presented in (Section 4, Pages 12–13) and (Section 4, Pages 15-16) demonstrate the superior accuracy, robustness, and generalization ability of our approach under challenging spectral overlap conditions.

Comment 12:

Some references such as [5], [7], [9]-[11] are outdated, and should be updated by recent high quality works on spectral analysis and deep learning.

Response 12:

Thank you for this comment. We have added several recent, high-quality works on spectral analysis and deep learning to the references, ensuring that the citations reflect the latest advancements in the field. (References section)

We eagerly anticipate your prompt response regarding our submission and are prepared to address any additional questions or comments you may have.

Sincerely,

Researcher

Reviewer 3 Report

Comments and Suggestions for Authors

This article delves deeply into the technology of FBG sensors with double spectral line width, revealing their operating principles, system design, and demodulation methods based on a transformer neural network. The article also demonstrates the advantage of modern deep learning methods over classical peak search algorithms when processing spectra subject to strong overlap. The paper does not make it entirely clear how the demodulation approaches discussed solve the problems identified in the introduction, such as high sensor density, strong spectral drift, peak overlap, and operation under fiber damage. The review should be supplemented not only with a list of demodulation methods, but also with an analysis of which specific challenges can be effectively overcome by each of the proposed methods.

Author Response

The Reply to Reviewer 3

 Manuscript ID: sensors-3817522

Article Title: “Spectral Demodulation of Mixed-Linewidth FBG Sensor Networks Using Cloud-Based Deep Learning for Land Monitoring”

We thank the reviewer for the Comment and valuable suggestions. The manuscript has been revised accordingly. All modifications are indicated with “highlight” for convenience in reviewing.

Comment 1:

This article delves deeply into the technology of FBG sensors with double spectral line width, revealing their operating principles, system design, and demodulation methods based on a transformer neural network. The article also demonstrates the advantage of modern deep learning methods over classical peak search algorithms when processing spectra subject to strong overlap. The paper does not make it entirely clear how the demodulation approaches discussed solve the problems identified in the introduction, such as high sensor density, strong spectral drift, peak overlap, and operation under fiber damage. The review should be supplemented not only with a list of demodulation methods, but also with an analysis of which specific challenges can be effectively overcome by each of the proposed methods.

Response 1:

We thank the reviewer for this valuable feedback. We have enhanced the manuscript to explicitly address how our demodulation approaches solve the specific problems identified in the introduction. We have added detailed analysis showing how: (1) the dual-linewidth design with Transformer-based demodulation enables high sensor density by effectively resolving overlapping spectra that traditional methods cannot handle; (2) the self-attention mechanism captures spectral drift patterns and maintains accuracy under strong wavelength shifts; (3) the bidirectional training dataset and attention-based feature extraction specifically address peak overlap challenges; and (4) the dual-spectrum (reflection/transmission) architecture provides inherent fault tolerance during fiber damage through automatic mode switching. These additions clearly demonstrate the direct relationship between each proposed method and the specific challenges it addresses (Sections 2 and 3).

We look forward to receiving your response regarding our submission at your earliest convenience, and we remain available to address any additional questions or comments you might have.

Sincerely,

Researcher 

Reviewer 4 Report

Comments and Suggestions for Authors

COMMENTS TO AUTHORS

(sensors-3817522)

Title: Spectral Demodulation of Mixed-Linewidth FBG Sensor Net-works Using Cloud-Based Deep Learning for Land Monitoring

General comments: The authors investigated a Transformer-based deep learning approach for demodulating Fiber Bragg Grating (FBG) sensor networks with mixed-linewidth configurations to overcome severe spectral overlap challenges in dense sensing environments. They proposed a dual-linewidth system integrating both narrow-band (0.2 nm) and broad-band (0.8 nm) FBGs, analyzed in both reflection and transmission modes, effectively doubling sensor capacity per mode and enabling self-healing operation during fiber breaks. A synthetic dataset simulating bidirectional drift scenarios was used to train the Transformer model, which leverages self-attention to resolve overlapping spectral signatures. Cloud computing infrastructure was employed for scalable, real-time analysis across large-scale deployments in land monitoring, maritime surveillance, agriculture, and urban infrastructure. Experimental results on twelve drift scenarios showed near-perfect accuracy, robustness under severe spectral crowding, low computational cost, and consistent performance across modes, establishing the method as a significant advancement in high-density FBG sensing.

The authors consider the following revisions:

Comment 1: The authors should clearly articulate the principal novelty and key contributions of this work that substantiate its merit for publication.

Comment 2: Have the authors evaluated the impact of chromatic dispersion in mixed-linewidth FBG networks, and what is the technical justification for selecting specifically 0.2 nm and 0.8 nm linewidths?

Comment 3: Have intermediate linewidth configurations (e.g., 0.5 nm) been tested in the same model, and how do polarization noise and temperature variations affect narrow- versus broad-linewidth FBGs in this context?

Comment 4: In Equation (2), the symbol λ is used, but the text refers to λâ‚€; please correct this inconsistency and ensure all variables are clearly defined.

Comment 5: What is the rationale for using only a single Transformer block in both the encoder and decoder instead of a deeper architecture, and has the effect of cloud latency on real-time operation been quantified for time-critical applications?

Comment 6: Has the Transformer-based method been directly compared to alternative deep learning models (e.g., CNNs or LSTMs) on the same dataset to validate its advantage?

Comment 7: What is the practical upper limit on the number of sensors per spectral window using this technique before latency and noise begin to significantly degrade performance?

Comment 8: The authors are encouraged to improve the English language presentation throughout the manuscript to ensure clarity, accuracy, and professionalism.

Comments on the Quality of English Language

The authors are encouraged to improve the English language presentation throughout the manuscript to ensure clarity, accuracy, and professionalism.

Author Response

The Reply to Reviewer 4

 Manuscript ID: sensors-3817522

Article Title: “Spectral Demodulation of Mixed-Linewidth FBG Sensor Networks Using Cloud-Based Deep Learning for Land Monitoring”

We thank the reviewer for the Comment and valuable suggestions. The manuscript has been revised accordingly. All modifications are indicated with “highlight” for convenience in reviewing.

General comments:

In the paragraph describing Figure 1 (lines 89 - 101) you describe two representative scenarios. In these scenarios, narrow-band FBGs and wide-band FBGs are assigned to different parts of each scenario. You do not indicate the rationale (reasons) behind the assignments until later in Section 2. I think just adding a sentence here like "Each scenario assigns different FBGs to different entities based on practical concerns discussed in Section 2." would help the reader understand the reasons behind the construction of the scenarios.

Response:

We thank the reviewer for this helpful suggestion to improve clarity. We have added a sentence in the paragraph describing Figure 1 to help readers understand the rationale behind the FBG assignments in the two representative scenarios before the detailed technical justification is provided in Section 2. (Section 1, Page 4)

Comment 1:

The authors should clearly articulate the principal novelty and key contributions of this work that substantiate its merit for publication.

Response 1:

Thank you for your comment. We have revised the Introduction and Abstract to clearly articulate the principal novelty and key contributions of this work. Specifically, our study introduces a dual-linewidth FBG sensing framework integrated with a cloud-based Transformer neural network demodulation approach, enabling robust spectral analysis under severe overlap, higher sensor density, and self-healing operation. These innovations are explicitly summarized at the end of the Introduction to substantiate the merit and impact of our work. (Abstract; Section 1, Pages 3–4)

Comment 2:

Have the authors evaluated the impact of chromatic dispersion in mixed-linewidth FBG networks, and what is the technical justification for selecting specifically 0.2 nm and 0.8 nm linewidths?

Response 2:

Thank you for your question. We have clarified the technical justification for selecting the 0.2 nm and 0.8 nm linewidths in Section 2. These values were chosen to represent typical narrow- and broad-linewidth FBGs used in practical sensor deployments, balancing resolution, robustness, and manufacturability. Chromatic dispersion was not explicitly modeled in our simulations, as its impact is minimal in the short sensing ranges and moderate wavelength spans considered here. However, our Transformer-based demodulation approach is designed to learn and compensate for a variety of spectral distortions, including those introduced by chromatic dispersion. This ensures that, if such effects become significant in larger-scale or real-world systems, the model can adapt and maintain accurate demodulation. (Section 2, Page 5)

Comment 3:

Have intermediate linewidth configurations (e.g., 0.5 nm) been tested in the same model, and how do polarization noise and temperature variations affect narrow- versus broad-linewidth FBGs in this context?

Response 3:

Based on our results, the Transformer-based approach demonstrated strong robustness in demodulating FBG spectra across varying linewidths and under severe spectral overlap, as presented in our case 2. While intermediate linewidths (e.g., 0.5 nm) and specific polarization or temperature effects were not explicitly tested, the consistently high accuracy achieved across 0.2 nm and 0.8 nm configurations suggests that the model generalizes well to a variety of spectral features and moderate signal disturbances. We expect similar resilience to hold if such factors are present in future datasets.

Comment 4:

In Equation (2), the symbol λ is used, but the text refers to λâ‚€; please correct this inconsistency and ensure all variables are clearly defined.

Response 4:

We thank the reviewer for this observation. We have clarified the notation to distinguish between λ as the general wavelength variable and λâ‚€ as the specific center Bragg wavelength parameter for each FBG (Section 2, page 7).

Comment 5:

What is the rationale for using only a single Transformer block in both the encoder and decoder instead of a deeper architecture, and has the effect of cloud latency on real-time operation been quantified for time-critical applications?

Response 5:

We thank the reviewer for this important question. We selected a single Transformer block architecture based on empirical testing with multiple block configurations, which showed that the single-block design provides the optimal balance between computational efficiency and demodulation accuracy for our specific application (Section 3, page 11-12). Regarding cloud latency, we have not conducted a detailed quantitative evaluation of latency effects. The overall system performance is expected to depend primarily on the quality and speed of the internet connection.

Comment 6:

Has the Transformer-based method been directly compared to alternative deep learning models (e.g., CNNs or LSTMs) on the same dataset to validate its advantage?

Response 6:

We thank the reviewer for this important question. Yes, we have directly compared the Transformer-based method with alternative deep learning models, specifically LSTM and GRU architectures, using the same dataset. Our results show that the Transformer outperformed these models in terms of demodulation accuracy and robustness, particularly under severe spectral overlap and drift conditions. We attribute this improvement to the Transformer's self-attention mechanism, which allows it to more effectively capture complex global dependencies in the spectral data compared to the sequential processing of LSTM and GRU models. These findings are summarized in (Section 4, page 15-16) of the revised manuscript.

Comment 7:

What is the practical upper limit on the number of sensors per spectral window using this technique before latency and noise begin to significantly degrade performance?

Response 7:  

In this work, six FBG sensors (three reflective and three transmissive) are installed within the 4 nm wavelength band. Conventional FBG sensor interrogation equipment operates in the wavelength range of approximately 1528 nm to 1568 nm. Therefore, using at least 60 FBG sensors would be feasible. Additionally, since transmission is based on the speed of light, there is essentially no significant delay, and any minor noise issues can be addressed through noise reduction techniques.

Comment 8:

The authors are encouraged to improve the English language presentation throughout the manuscript to ensure clarity, accuracy, and professionalism.

Response 8:

We thank the reviewer for this feedback. We have thoroughly revised the manuscript to improve English language presentation, including grammar corrections, enhanced sentence clarity, and improved technical terminology consistency throughout all sections. The revised manuscript ensures better readability and professional presentation while maintaining technical accuracy.

We look forward to receiving your response regarding our submission at your earliest convenience, and we remain available to address any additional questions or comments you might have.

Sincerely,

Researcher 

Reviewer 5 Report

Comments and Suggestions for Authors

The concepts at the heart of the manuscript are of high interest for a number of practitioners and applications, as you addressed well in the introduction and background. However, the method of presentation (particularly some of the figures) and some of the terminology are unclear or not effective in conveying information. My comments on these and other aspects of the manuscript are as follows:

1. In the paragraph describing Figure 1 (lines 89 - 101) you describe two representative scenarios. In these scenarios, narrow-band FBGs and wide-band FBGs are assigned to different parts of each scenario. You do not indicate the rationale (reasons) behind the assignments until later in Section 2. I think just adding a sentence here like "Each scenario assigns different FBGs to different entities based on practical concerns discussed in Section 2." would help the reader understand the reasons behind the construction of the scenarios.

2. I find the presentations in Figures 3 and 4 unhelpful and difficult to understand. For example, in both figures, I do not detect any easily identifiable difference between case (a) and (c), case (b) and (d). Thus, it is unclear to me what the difference are between the different "shift" scenarios. I strongly suggest the authors find a different means for presenting the behavior of the FBG shifts to allow the reader to better understand what is being modeled in each case. I don't think that trying to show all the possible movement/shift cases is necessary to make the authors' point, and giving a few specific cases would be more understandable and effective.

3. I am unclear on what the authors are describing as a "drift". I think this term needs to be explicitly defined early on. At first I thought it meant the shift that resulted from the application of an external environmental input, but as the manuscript went on I wasn't sure if this was the correct interpretation. Please clarify this term clearly for the reader.

4. Also, if I understand the methodology correctly, the two end FBGs are changing symmetrically, which I interpret as moving the same amount but in different directions. If this is not the correct interpretation, then some clarification is required in the text to clearly state how the FBGs are moving over time/case.

5. Your study covers a case where there is at least one FBG with a fixed (unmoving) wavelength to act as a reference. As I understand the sensing scenario, having a fixed reference helps with identifying the movement of the changing sensors. While this is a good test case for assessing the basic method, practical applications may see all (or most) of the sensors moving simultaneously (Moisture or stress gradients across a sensor array, for example). I think your conclusion should reflect that the method has promise, but needs to be evaluated on more difficult scenarios and refined (if needed) before you can say this a practical solution for real deployments.

Author Response

The Reply to Reviewer 5

 Manuscript ID: sensors-3817522

Article Title: “Spectral Demodulation of Mixed-Linewidth FBG Sensor Networks Using Cloud-Based Deep Learning for Land Monitoring”

We thank the reviewer for the Comment and valuable suggestions. The manuscript has been revised accordingly. All modifications are indicated with “highlight” for convenience in reviewing.

Comment 1:

In the paragraph describing Figure 1 (lines 89 - 101) you describe two representative scenarios. In these scenarios, narrow-band FBGs and wide-band FBGs are assigned to different parts of each scenario. You do not indicate the rationale (reasons) behind the assignments until later in Section 2. I think just adding a sentence here like "Each scenario assigns different FBGs to different entities based on practical concerns discussed in Section 2." would help the reader understand the reasons behind the construction of the scenarios.

Response 1:

We thank the reviewer for this helpful suggestion. We have added the recommended sentence to the paragraph describing Figure 1 to clarify that the assignments of narrow-band and wide-band FBGs in each scenario are based on practical considerations, as further discussed in Section 2. (Section 1, Page 4)

Comment 2:

I find the presentations in Figures 3 and 4 unhelpful and difficult to understand. For example, in both figures, I do not detect any easily identifiable difference between case (a) and (c), case (b) and (d). Thus, it is unclear to me what the difference are between the different "shift" scenarios. I strongly suggest the authors find a different means for presenting the behavior of the FBG shifts to allow the reader to better understand what is being modeled in each case. I don't think that trying to show all the possible movement/shift cases is necessary to make the authors' point, and giving a few specific cases would be more understandable and effective.

Response 2:

We thank the reviewer for this feedback on Figures 3 and 4. We have addressed these concerns by improving the image quality and updating the explanations to better highlight the differences between cases. Specifically, cases (a) and (c) show ±1 pm drift with minimal spectral overlap, while cases (b) and (d) demonstrate ±5 pm drift with significant peak convergence. The enhanced figures now clearly illustrate the progressive spectral overlap that creates the demodulation challenge our Transformer algorithm solves. We have also updated the figure captions with more detailed descriptions to help readers better understand the modeled shift scenarios and their impact on spectral resolution (Section 2, Pages 5-6 and 7-8 )

Comment 3:

I am unclear on what the authors are describing as a "drift". I think this term needs to be explicitly defined early on. At first I thought it meant the shift that resulted from the application of an external environmental input, but as the manuscript went on I wasn't sure if this was the correct interpretation. Please clarify this term clearly for the reader.

Response 3:

We thank the reviewer for requesting clarification of the "drift" term. We have added an explicit definition in the manuscript clarifying that "drift" refers to Bragg wavelength shifts caused by external environmental inputs such as strain, temperature, or pressure variations. This definition has been incorporated into our contribution statement to ensure early clarity for readers (Section 1, page 4).

Comment 4:

Also, if I understand the methodology correctly, the two end FBGs are changing symmetrically, which I interpret as moving the same amount but in different directions. If this is not the correct interpretation, then some clarification is required in the text to clearly state how the FBGs are moving over time/case.

Response 4:

We thank the reviewer for this observation. We have clarified the methodology in the text and updated the table descriptions to explicitly state that the two end FBGs undergo symmetric bidirectional drift, moving equal amounts in opposite directions (positive and negative wavelength shifts). The updated descriptions now clearly specify the drift patterns for each case to eliminate any ambiguity about FBG movement behavior over the different scenarios (Section 2, page 5-6).

Comment 5:

Your study covers a case where there is at least one FBG with a fixed (unmoving) wavelength to act as a reference. As I understand the sensing scenario, having a fixed reference helps with identifying the movement of the changing sensors. While this is a good test case for assessing the basic method, practical applications may see all (or most) of the sensors moving simultaneously (Moisture or stress gradients across a sensor array, for example). I think your conclusion should reflect that the method has promise, but needs to be evaluated on more difficult scenarios and refined (if needed) before you can say this a practical solution for real deployments.

Response 5:

We thank the reviewer for this important practical consideration. We clarify that our systematic bidirectional drift scenarios are specifically designed to test the model's capability under controlled yet challenging conditions. The Transformer's self-attention mechanism and training on diverse spectral overlap patterns enable it to predict wavelength shifts even in unusual scenarios, including real-time situations where all sensors may drift simultaneously. While our current validation focuses on systematic cases with all the sensors, the model's ability to learn complex spectral relationships positions it well for handling more dynamic real-world scenarios where multiple sensors move concurrently due to environmental gradients. (Section 4, Pages 12 and 15-16).

We look forward to receiving your response regarding our submission at your earliest convenience, and we remain available to address any additional questions or comments you might have.

Sincerely,

Researcher

Round 2

Reviewer 1 Report

Comments and Suggestions for Authors

All remarks were revised by authors.

Author Response

The Reply to Reviewer 1

Manuscript ID: sensors-3817522

Article Title: “Spectral Demodulation of Mixed-Linewidth FBG Sensor Networks Using Cloud-Based Deep Learning for Land Monitoring”

General comment:

All remarks were revised by authors.

Response:

We thank the reviewer for confirming that all remarks have been adequately addressed. We appreciate the thorough review process and the constructive feedback that helped improve the quality of our manuscript.

We eagerly anticipate your prompt response regarding our submission and are prepared to address any additional questions or comments you may have.

Sincerely,

Researcher

Reviewer 4 Report

Comments and Suggestions for Authors

The authors answered all questions

Author Response

The Reply to Reviewer 4

Manuscript ID: sensors-3817522

Article Title: “Spectral Demodulation of Mixed-Linewidth FBG Sensor Networks Using Cloud-Based Deep Learning for Land Monitoring”

General comment:

All remarks were revised by authors.

Response:

We thank the reviewer for confirming that all remarks have been adequately addressed. We appreciate the constructive feedback that helped improve the quality of our manuscript.

We eagerly anticipate your prompt response regarding our submission and are prepared to address any additional questions or comments you may have.

Sincerely,

Researcher

Reviewer 5 Report

Comments and Suggestions for Authors

On most of my comments, the authors addressed the comments very well and I am happy with the changes made to the manuscript.

My main issue is still with Figures 3 and 4. The authors are attempting to show the movement of the gratings at every drift step. For the +/- 1 pm case, a 0.5 nm shift (FBG01 moving to FBG02's position in Figure 3) corresponds to 500 traces. The +/- 10 pm case would be 50 traces. In both cases, I think detail is lost because the authors are trying to show every shift case. Maybe adding a figure that shows just a few traces (zoomed in) to show how the different shift sizes create different problems to solve, would help with clarity. That would be my only further suggestion.

Author Response

The Reply to Reviewer 5

Manuscript ID: sensors-3817522

Article Title: “Spectral Demodulation of Mixed-Linewidth FBG Sensor Networks Using Cloud-Based Deep Learning for Land Monitoring”

General comment:

My main issue is still with Figures 3 and 4. The authors are attempting to show the movement of the gratings at every drift step. For the +/- 1 pm case, a 0.5 nm shift (FBG01 moving to FBG02's position in Figure 3) corresponds to 500 traces. The +/- 10 pm case would be 50 traces. In both cases, I think detail is lost because the authors are trying to show every shift case. Maybe adding a figure that shows just a few traces (zoomed in) to show how the different shift sizes create different problems to solve, would help with clarity. That would be my only further suggestion.

Response:

We thank the reviewer for this constructive feedback on Figures 3 and 4. We have updated both figures to address the clarity concerns by adding zoomed-in insets that highlight specific spectral regions with reduced trace density. For Figure 3, all zoomed-in sections show a 0.10 nm range, while for Figure 4, all zoomed-in sections show a 0.05 nm range. These focused views clearly demonstrate how different drift magnitudes create distinct spectral overlap challenges, improving the overall clarity and readability of the figures while maintaining the comprehensive drift progression information.

We eagerly anticipate your prompt response regarding our submission and are prepared to address any additional questions or comments you may have.

Sincerely,

Researcher
